# Polycaprolactone (PCL)-Polylactic Acid (PLA)-Glycerol (Gly) Composites Incorporated with Zinc Oxide Nanoparticles (ZnO-NPs) and Tea Tree Essential Oil (TTEO) for Tissue Engineering Applications

**DOI:** 10.3390/pharmaceutics15010043

**Published:** 2022-12-22

**Authors:** Carlos David Grande-Tovar, Jorge Iván Castro, Carlos Humberto Valencia Llano, Diego López Tenorio, Marcela Saavedra, Paula A. Zapata, Manuel N. Chaur

**Affiliations:** 1Grupo de Investigación de Fotoquímica y Fotobiología, Universidad del Atlántico, Carrera 30 Número 8-49, Puerto Colombia 081008, Colombia; 2Laboratorio SIMERQO, Departamento de Química, Universidad del Valle, Calle 13 No. 100-00, Cali 76001, Colombia; 3Grupo Biomateriales Dentales, Escuela de Odontología, Universidad del Valle, Calle 4B # 36-00, Cali 76001, Colombia; 4Grupo de Polímeros, Facultad de Química y Biología, Universidad de Santiago de Chile, USACH, Santiago 9170020, Chile

**Keywords:** biocompatibility, zinc oxide nanoparticles, tissue engineering, nanocomposites

## Abstract

The search for new biocompatible materials that can replace invasive materials in biomedical applications has increased due to the great demand derived from accidents and diseases such as cancer in various tissues. In this sense, four formulations based on polycaprolactone (PCL) and polylactic acid (PLA) incorporated with zinc oxide nanoparticles (ZnO-NPs) and tea tree essential oil (TTEO) were prepared. The sol-gel method was used for zinc oxide nanoparticle synthesis with an average size of 11 ± 2 nm and spherical morphology. On the other hand, Fourier Transformed infrared spectroscopy (FTIR) showed characteristic functional groups for each composite component. The TTEO incorporation in the formulations was related to the increased intensity of the C-O-C band. The thermal properties of the materials show that the degradative properties of the ZnO-NPs decrease the thermal stability. The morphological study by scanning electron microscopy (SEM) showed that the presence of TTEO and ZnO-NPs act synergistically, obtaining smooth surfaces, whereas membranes with the presence of ZnO-NPs or TTEO only show porous morphologies. Histological implantation of the membranes showed biocompatibility and biodegradability after 60 days of implantation. This degradation occurs through the fragmentation of the larger particles with the presence of connective tissue constituted by type III collagen fibers, blood vessels, and inflammatory cells, where the process of resorption of the implanted material continues.

## 1. Introduction

The use of synthetic polymers such as PLA (polylactic acid) or PCL (polycaprolactone) is widespread in food packaging, mainly due to their low cost, thermal properties, resistance to UV degradation, and high mechanical properties [1,2]. Additionally, thanks to their biodegradable and biocompatible properties, they have been explored in the field of tissue engineering [3,4].

PLA is a semi-crystalline and synthetic polymer from lactic acid derived from carbohydrate fermentation [5,6]. On the other hand, PCL is a thermoplastic polymer from ε-caprolactone polymerized by a catalyst-induced ring-opening mechanism [7]. However, these polymers present an intrinsic stiffness of the carbonated chain, which becomes a disadvantage due to their low tensile elongation and toughness properties that limit their industrial applications [8]. The addition of plasticizers (essential oils) or nanofillers (inorganic nanoparticles) has been proposed to overcome these low elongation and toughness capacities.

Zinc oxide nanoparticles (ZnO-NPs) are a multifunctional material recognized as safe materials used as a food additive generally regarded as safe (GRAS) and approved by the Food and Drug Administration (FDA) [9]. They are excellent and low-cost semiconductors with good chemical stability, biocompatibility, and antimicrobial activity against a wide range of microorganisms such as *Escherichia coli*, *Pseudomonas aeruginosa*, and *Staphylococcus aureus* [10,11], becoming attractive for several applications on textiles [12], packaging [13], and medicine [14]. Several methods to evaluate the influence of ZnO-NPs in polymer composites have been studied, including the formation of PLA/ZnO nanocomposites, which showed antimicrobial activity against bacteria such as *E. coli*, *S. aureus*, *Klebsiella pneumoniae*, among others [1,15,16,17].

Essential oils are constituted by volatile compounds that improve physical and structural properties in polymeric matrices. Additionally, due to the organic structural constitution of the oil, they provide antimicrobial properties. *Melaleuca alternifolia*, or tea tree essential oil (TTEO), contains monoterpenes, sesquiterpenes, and alcohol derivatives [18]. The main compounds in TTEO are terpene-4-ol (≥30%), γ-terpene (about 20%), α-terpinene (about 8%), *p*-cymene (about 8%), α-pinene (about 3%), terpinolene (about 3%), and 1,8-cineol (≤15%) [19]. Previous studies have shown that TTEO possesses anti-inflammatory and antimicrobial properties. However, the direct application of TTEO to food is limited due to the hydrophobic naturalization and the astringent flavor intensity [20].

The antimicrobial properties of TTEO are well documented for the control of yeasts and bacteria. In this regard, TTEO showed antimicrobial activity when applied on ground beef against *Listeria monocytogenes* at a concentration of 1.5% *w/v* [21]. Likewise, chitosan membranes enriched with TTEO showed 100% inhibition of *L. monocytogenes* growth [22].

Our group previously prepared four chitosan (CS)/TTEO/polyvinyl alcohol (PVA) membranes, where TTEO acted as a plasticizer. This study found that the addition of TTEO increased biodegradability and biocompatibility due to decreased inflammatory infiltrate and almost total bioabsorption after 90 days of implantation [23]. However, to our knowledge, there are no studies involving the biocompatibility evaluation in vivo of zinc oxide nanoparticles and TTEO within a PLA/PCL polymeric matrix, which could improve the biological response and the stability of the materials. For this reason, we assessed the preliminary biocompatibility of PCL/PLA/ZnO-NPs/Gly/TTEO membranes based on subdermal implantation in Wistar rats. The results of increased and accelerated resorption and the absence of an aggressive inflammatory response demonstrated the potential application of these materials in tissue engineering.

## 2. Materials and Methods

### 2.1. Materials

All the reagents used in this investigation were obtained from commercial distributors. The PLA was obtained from NatureWorks (Minnetonka, MN, USA) with approximately 2% of the D isomer with a molecular weight of 200,000 Da. The PCL used was 6800 layers obtained from Perstorp Company (Warrington, UK) which contained a molecular weight of 80,000 g/mol with a flow rate of 3 g/10 min. For the formation of ZnO nanoparticles, 99% sodium hydroxide, 80–100% pure zinc chloride, and 99.8% 2-propanol were obtained from Merck (Burlington, MA, USA).

### 2.2. Synthesis of ZnO-NPs

The synthesis of the nanoparticles was carried out according to the procedure already reported [24]. Initially, two solutions were used: the first consisted of 5.5 g of ZnCl_2_ in 200 mL of distilled water. The second solution was a 5 M solution of NaOH in distilled water. Subsequently, the second solution was added over the first dropwise, maintaining the temperature at 90 °C for 10 min. Next, the obtained suspension was washed with abundant distilled water to decrease the concentration of NaCl present. Then, the nanoparticles were dispersed in an ultrasonication bath (Branson, Madrid, Spain) and mixed with 2-propanol for 10 min at 20 °C. Following this, the ZnO-NPs were centrifugated at 5000 rpm for 15 min by re-washing with 2-propanol in triplicate. Finally, the nanoparticles were calcined in an oven Nabertherm LHT 02/18 (Lilienthal, Bremen, Germany) at 250 °C for five hours [10].

### 2.3. Tea Tree Essential Oil Composition (TTEO)

The composition and study of the TTEO components were carried out according to previous works of our group [23]. The essential tea oil (TTEO) was purchased from Marnys (Madrid, Spain), and its compositional analysis was characterized by gas chromatography coupled to mass spectrometry (GC-MS) using C6 to C25 hydrocarbons as reference. The gas chromatograph was an AT6890 series plus (Agilent Technologies, Palo Alto, CA, USA) coupled to a mass selective detector (Agilent Technologies, MSD 5975). Column DB-5MS (J & W Scientific, Folsom, CA, USA), 5% -Ph-PDMS. Identification comparing RI (retention indexes) with the Adams database (Wiley, 138 and NIST05, Agilent, Santa Clara, CA, USA).

### 2.4. Preparation of the Membranes of PCL/PLA/ZnO-NPs/Gly/TTEO

For the preparation of the PCL/PLA/ZnO-NPs/Gly/TTEO membranes, the main parameter considered was the presence of a concentration final of 4% (wt.%), according to the wt.% ratio of each formulation. Then, a dispersion of ZnO-NPs (300 mg/10 mL) in chloroform was prepared using an ultrasonic bath (Branson, Madrid, Spain) for two hours. The glycerol used had a density of 1.26 g/cm^3^. Subsequently, each component was dissolved in chloroform and mixed according to Table 1. Finally, the resulting mixture was placed in an ultrasonic bath (Branson, Madrid, Spain) for two hours to eliminate the pre-existing bubbles in the solution.

The mixture was poured into glass molds for 24 h and cured in a preheated oven at 40 °C ± 0.2 to obtain PCL/PLA/ZnO-NPs/Gly/TTEO membranes.

### 2.5. Characterization of ZnO-NPs

The morphology of the ZnO-NPs was measured by transmission electron microscopy (TEM) using a JEOL ARM 200 F (Tokyo, Japan) equipment operating at 20 kV. The sample preparation included a drop of ZnO-NPs mixed with ethanol and sonicated on a standard carbon-coated copper grid using a 400 mesh screen. The average of 100 zinc oxide nanoparticles was considered to determine the particle size by processing in Image J 1.49q software. The crystalline structure was evaluated using a PANalytical X0Pert PRO diffractometer (Malvern Panalytical, Jarman Way, Royston, UK) using copper radiation with a wavelength of Kα1 (1.540598 Å) and Kα2 (1.544426 Å) operated in the secondary electron mode at 45 kV in a 2θ range between 5 and 70°. Finally, the chemical structure of the nanoparticles was performed by Fourier-transform infrared spectroscopy on an FT-IR-8400 (Shimadzu, Kyoto, Japan) in a wavenumber range between 500–4000 cm^−1^.

### 2.6. Characterization of PCL/PLA/ZnO-NPs/Gly/TTEO Nanocomposites

#### 2.6.1. Fourier Transform Infrared Spectroscopy

The membranes’ functional groups were determined by an FT-IR-8400 spectrophotometer (Shimadzu, Kyoto, Japan) using a diamond tip as an accessory in a wavenumber range between 500–4000 cm^−1^.

#### 2.6.2. X-ray Diffraction

The diffraction planes for the nanocomposites were determined under the same conditions as the ZnO-NPs were.

#### 2.6.3. Scanning Electron Microscopy (SEM)

Surface morphology of PCL/PLA/ZnO-NPs/Gly/TTEO membranes was performed through a Hitachi TM 3000 scanning electron microscope (Musashino, Tokyo, Japan) in the secondary electron mode using a 20 kV voltage accelerator with a gold layer to increase the conductivity of the samples.

#### 2.6.4. Thermal Analysis of PCL/PLA/ZnO-NPs/Gly/TTEO Membranes

The thermal properties of the PCL/PLA/Gly/ZnO-NPs/TTEO membranes were measured by thermogravimetric analysis (TGA) on a NETZSCH TG 209 F1 Libra (Mettler Toledo, Schwerzenbach, Switzerland). Each nanocomposite was introduced on an alumina-based microanalytical balance and heated in a range of 25–900 °C at a temperature flow rate of 10 °C/min (nitrogen atmosphere at a rate of 50 mL/min). Other properties such as glass transition temperature (*T_g_*), crystallization temperature (*T_c_*), melting temperature (*T_m_*), enthalpy of crystallization (Δ$H_{cc})$, and enthalpy of fusion (Δ$H_{m})$ were determined by differential scanning calorimetry (DSC) on a DSC1/500 (Mettler Toledo, Schwerzenbach, Switzerland). The membranes were scanned from 25–250 °C at a rate of 10 °C/min and a 60 mL/min nitrogen flow rate. Both TGA and DSC data were analyzed through the instrument software using TA instruments Universal Analysis Software 2000 version 4.5A.

The polymers’ percentage crystallinity was determined by Equation (1) [25].
(1)$$X_{c}=\frac{\left(\Delta H_{m}-\Delta H_{CC}\right)}{\Delta H_{m}^{{}^{\circ}}\left(1-x\right)}$$
where $\Delta H_{m}^{{}^{\circ}}$ corresponds to the theoretical enthalpy of the completely crystalline polymer, which is 93 J/g for PLA [26]. $\Delta H_{m}$ and $\Delta H_{CC}$ are the enthalpy of fusion and enthalpy of crystallization of the nanocomposites in J/g, and the wt.% of PLA is given by 1 − x term.

### 2.7. In Vivo Biocompatibility Study of PCL/PLA/ZnO-NPs/Gly/TTEO Membranes

#### 2.7.1. Surgical Preparation of Biomodels

For the surgical preparation of the biomodels, the recommendations of UNE: 10993-6 (Biological evaluation of medical devices—Part 6: Tests for local effects after implantation. ISO 10993-6:1994). Three four-month-old male Wistar rats (Rattus norvegicus domestic) with an average weight of 380 g were randomly selected from the vivarium population and subjected to pocket incisions where the different membranes were implanted.

The biomodels were sedated by intramuscular application of a solution of Ketamine 70 mg/kg (Blaskov Laboratory, Bogotá, Colombia) and Xylazine 30 mg/kg (ERMA Laboratories, Celta, Colombia). Subsequently, they were shaved to remove the hair on the dorsal surface, and the area was disinfected with an isodine solution (Laboratory Sanfer, Bogotá, Colombia). Then, the area to be implanted was determined (right side of the midline), where anesthesia was applied with an infiltrative technique using 2% Lidocaine with epinephrine (Newstetic, Guarne, Colombia). Then, four 5 mm incisions were made on the right side of the midline, and four pockets 10 mm wide and 10 mm deep were created with hemostatic forceps, with a centimeter of separation between them [27].

Two months after implantation of the biomodels, euthanasia procedures were performed by intramuscular injection of sodium pentobarbital/sodium diphenylhydantoin (0.3 mL/biomodel kg) (Euthanex, INVET Laboratory, Cota, Colombia). Samples were recovered, fixed with buffered formalin for 48 h, washed with phosphate-buffered saline at neutral pH (PBS), and dehydrated by immersion in 70 to 100% ascending alcohol. Finally, the inclusion procedures were performed by diaphanization with Xylol and infiltration with kerosene using the Auto-technicon Tissue Processor equipment. ™(Leica Microsystems, Mannheim, Germany), and kerosene blocks were obtained for histological processing with Thermo Scientific equipment. ™Histoplast Paraffin™ kit (Fisher Scientific. Waltham, MA, USA).

#### 2.7.2. Histology Analysis

After obtaining the kerosene blocks, they were cut into 5 µm sections with a Leica microtome (Leica Microsystems, Mannheim, Germany), and the sections were deposited on glass slides for histological processing. The samples were then analyzed after 48 h with hematoxylin-eosin (HE) and Gomori trichrome (GT) staining. Histological images were taken using a Leica DM750 optical microscope and a Leica DFC 295 camera. The images were processed by Leica Application Suite version 4.12.0 software (Leica Microsystem, Mannheim, Germany).

The procedures were carried out according to the ARRIVE (Animal Research: Reporting of In Vivo Experiments) guidelines [28]. The number of animals was determined according to the UNE-EN-30993-6 standard, equivalent to the ISO 10993-6 standard [27]. The animals were supplied by the LABBIO laboratory (vivarium) of the Universidad del Valle in Cali, Colombia, and remained there for the duration of the research; the ethical endorsement and supervision were performed by the ethics committee with biomedical experimentation animals (CEAS) of the Universidad del Valle, through Resolution CEAS 006-022; There were no deaths in the intervened biomodels or intraoperative or post-surgical complications.

## 3. Results and Discussion

### 3.1. Tea Tree Essential Oil Characterization

According to the GC-MS analysis of TTEO (Appendix A), 52 compounds could be observed divided into nine monoterpenes, ten hydrocarbons, 27 oxygenated sesquiterpenes, five mixed compounds, and one unidentified compound, which were previously identified [23]. The compounds corresponding to terpinene-4-ol (≥30%) as the primary component, followed by γ-terpinene (about 20%), α-terpinene (about 8%), *p*-cymene (about 8%), α-pinene (about 3%), terpinolene (about 3%), among others [19,29], consistent with previous works [29].

### 3.2. Characterization of the ZnO-NPs

#### 3.2.1. Fourier Transform Infrared Spectroscopy (FT-IR)

Figure 1 shows the FT-IR spectra of the ZnO-NPs. The symmetric strain band at 3496 cm^−1^ corresponds to the O-H vibrational mode of the hydroxyl group. In addition, ZnO shows a signal from 560 cm^−1^, correlating to the vibrations of the Zn–O bond of the nanoparticle. The second signal, at 720 cm^−1^ and 900 cm^−1^, is attributed to the stretching and deformation vibrations of Zn–O [30].

#### 3.2.2. X-ray Diffraction (XRD) of ZnO-NPs

X-ray diffraction (XRD) studies the crystalline ZnO-NPs phase. The point group and hexagonal Wurtzite structure were confirmed due to the position and intensity of its peaks and the absence of impurities (Figure 2). There are evident angular displacements 2θ: 31.7, 34.4, 36.2, 47.5, 56.5, 62.8, 66.3, 67.8, and 68.9° according to the (100), (002), (101), (102), (110), (103), (200), (112), and (201) [31].

The average crystallite sizes (*τ*) for the ZnO-NPs were determined using the Debye-Scherrer Equation (2).
(2)$$\tau =\frac{K\lambda }{\beta Cos\left(\theta \right)}$$
where *K* is Scherrer constant, and the crystalline shape factor is 0.89, *λ* represents the wavelength of X-ray source 1.5405 Å used in XRD, *β* is full width at half maximum of diffraction peak, and *θ* is the Bragg angle of the intense peak. In this regard, from the diffractogram data of the ZnO-NPs, the average crystallite size calculated is 30.13 nm.

#### 3.2.3. Thermal Analysis for the ZnO-NPs

The thermal properties of the ZnO-NPs were determined by thermogravimetric analysis (TGA). TGA exhibits the mass loss (in percentage) with the degradation temperature of the nanocomposites. Figure 3 shows the thermogram with two stages of degradation and two clear degradation stages. The first degradation stage, with its maximum peak at 165 °C (1.8%), is attributed to water absorption and the presence of the hydroxyl groups from ZnO nanoparticles. The second degradation stage at 408 °C (1.4%) probably corresponds to the complete decomposition of ZnO after calcination, indicating that the zinc oxide is thermally stable [32].

#### 3.2.4. Transmission Electron Microscopy (TEM)

Figure 4 shows the morphology and size of the ZnO-NPs by transmission electron microscopy (TEM). Nearly spherical and fairly monodisperse nanoparticles are observed. However, some aggregates are found because when dry nanoparticle powder is added to water, the nanoparticles have a high local concentration with high surface energy, causing a high collision frequency and high aggregation [24,33]. The average diameter of the nanoparticles was found by averaging the size of 100 particles using Image J software. Considering the processed dimensions, we made a histogram that relates the frequency to the particle distribution (Figure 4B). In this sense, the diameter of the nanoparticles was 11 ± 2 nm.

### 3.3. FT-IR from PCL/PLA/ZnO-NPs/Gly/TTEO Nanocomposites

Figure 5 shows the characteristic vibrations of the PCL/PLA/ZnO-NPs/Gly/TTEO functional groups, which are highly sensitive to the structural composition of the scaffold.

In general, the formulations present symmetric tension bands at 3338 cm^−1^ attributed to the -OH group, the symmetric and asymmetric tension band of the alkyl groups -CH and -CH_2_ at 2947 cm^−1^, two tension bands corresponding to the C=O group at 1756 and 1722 cm^−1^ belonging to the carbonyl (ester) groups of both PLA and PCL, the asymmetric tension band of the aliphatic C-O-C group at 1182 cm^−1^ (for both PCL and PLA), and finally, the carbonate chain stretching band between 960–830 cm^−1^ present for both PCL and PLA. Concerning the formulations F2, F3, and F4, the band corresponding to the vibrational mode of ZnO at 682 cm^−1^ was not observable, probably due to the small amount of weight found in the formulations. Additionally, the band at 1456 cm^−1^ corresponds to the TTEO olefinic groups of the components (C=C) [34].

On the other hand, two characteristic bands are observed for F2 and F3 in the region of the carbonyl groups (1756 and 1724 cm^−1^). F4 only exhibited one carbonyl peak (1756 cm^−1^). In addition, there was an increase in the ester group intensity (1085 cm^−1^). This increase might correspond to the presence of the essential oil and the formation of hydrogen bonds with the polymers [35]. It may also be due to the accumulation of lactide molecules from the oligomeric ring and acetaldehyde groups resulting from the degradation of the chains by the presence of ZnO-NPs [16].

### 3.4. Thermal Analysis of PCL/PLA/ZnO-NPs/Gly/TTEO Nanocomposites

The thermal degradation for the PCL/PLA/ZnO-NPs/Gly/TTEO nanocomposites with temperature is observed in Figure 6. As previously mentioned, little thermal stability is observed within the thermograms due to the lack of intermolecular interaction between the polymers [36]. In addition, the degradative processes driven by ZnO-NPs introduction induce the formation of acetaldehyde, as previously reported [37].

Figure 6A shows the thermograms for F1, F2, F3 and F4. The thermal decomposition for formulation F1 (25%PCL/70%PLA/5%Gly) shows three stages of thermal degradation. The first degradation stage has a maximum decomposition peak at 174 °C (4.7%) corresponding to residual water molecules and low molecular weight molecules such as glycerol in the polymer matrix. The second degradation stage has its maximum degradation at 316 °C (28.4%) attributed to intramolecular transesterification forming lactide and cyclic oligomers. In addition, there is acrylic acid formation by the cis-elimination of carbon oxides and acetaldehyde, causing the fragmentation of the PLA chain [5]. Finally, the third degradation stage has a maximum peak of 400 °C (11.7%). The degradation is attributed to the PCL scission of hydroxyl groups followed by depolymerization [7].

On the other hand, the F2 membrane presents four degradation stages, whereas F3 and F4 present the same three degradation stages. The new degradation peak for the F2 membrane (22%PCL/70%PLA/5%Gly/3%ZnO-NPs) is probably due to zinc products intensifying the polymer matrix degradative processes [38,39], producing low molecular weight PLA and PCL through intermolecular transesterification and depolymerization below 337 °C [17]. Concerning the F3 membrane (15%PCL/70%PLA/5%Gly/10%TTEO), it was possible to observe the increase in thermal stability due to the presence of thermostable molecules from the essential oil of tea (TTEO). In addition, the low broadening of degradation stages is related to the increase in the hydrophobic character due to the incorporation of the essential oil.

Finally, for the F4 membrane (12%PCL/70%PLA/5%Gly/3%ZnO-NPs/10%TTEO), the degradable properties of ZnO predominate over the thermostable properties of the essential oil TTEO due to the decrease in the degradation peaks concerning the F3 membrane [40,41].

Differential scanning calorimetry (DSC) allows the study of the membranes’ different thermal properties and transition phases, as shown in Figure 7. This thermogram shows the melting temperatures of PCL (*T_m1_*) and PLA (*T_m2_*) and the glass transition temperatures (*T_g_*). Table 2 summarizes the above properties and the percentage of PLA crystallinity for each of the membranes.

The glass transition temperature for PLA is approximately 60 °C, close to the melting temperature of PCL (*T_m3_*). Therefore, the study of the influence of ZnO-NPs on the material’s crystallinity becomes very complex. The thermal characterization reveals that with the ZnO-NPs incorporation, a single packing of the PLA atoms is favored because the β form (orthorhombic or trigonal) is not found in the F2 and F4 formulations. In contrast, the α form (pseudo-orthorhombic, pseudo-hexagonal and orthorhombic) excels in all four membranes at 146 °C. These observations suggest a further transformation of the crystal structure from β to α form by the degradative action of ZnO-NPs [42]; moreover, the increase in the crystallization temperature for the F2 and F4 formulations is related to the rise of nucleation sites and the low crystallization rate of PLA.

Additionally, an increase in crystallization temperature was observed for PCL at 64 °C, suggesting that the degree of degradation caused by the ZnO-NPs predominates over the thermal properties of the membranes, leading to increased mobility of the chains toward the PCL nucleation sites.

### 3.5. XRD of PCL/PLA/ZnO-NPs/Gly/TTEO Nanocomposites

The interaction between the X-rays with the membranes to give characteristic diffraction planes for each material can be seen in Figure 8. The crystallographic data demonstrate the orthorhombic nature of the PLA with diffraction planes 2θ at 14.8, 16.7, 19.1, 22.2, and 29.2° attributed to planes (001), (101), (012), (013), and (004), respectively. It was also possible to observe the diffraction peaks of orthorhombic PCL at 21.4 and 23.7°, corresponding to the (110) and (200) planes. According to the above, the behavior of the polymers (PCL and PLA) inside the membranes preserves their crystallographic nature because they present the same displacement of the diffraction angle 2θ [43].

On the other hand, changes in the crystallinity of the F2, F3, and F4 membranes were observed. Firstly, adding ZnO-NPs promotes degradation by decreasing the nucleation processes, generating greater mobility of the polymer chains, and causing new diffraction sites such as the 2θ shift at 9° [44]. Secondly, the introduction of TTEO caused a decrease in the resolution and intensity of the diffraction peaks due to a change in the crystallinity; this observation suggests that TTEO reduces the intramolecular hydrogen bonding interactions generating intermolecular spaces along the polymer chains [43].

### 3.6. Scanning Electron Microscopy (SEM) of PCL/PLA/ZnO-NPs/Gly/TTEO Nanocomposites

Figure 9 shows the electron micrographs of the different membranes. The surface micrographs of the membranes show a porous drop-matrix structure that allows interaction with new components due to the incompatibility between the phases, especially F1. Additionally, as observed in previous works, the F2 formulation presents an increase in pore size, probably due to inadequate interactions between the nanoparticle interface and the polymeric matrix [45].

The irregularities observed in F3 as islands and microvoids are mainly due to the hydrophobic character of the lipids in the TTEO, causing flocculation of the components above the polymeric matrix. On the other hand, when the mixture is subjected to 40 °C in the preheated oven for membrane formation, the natural components of the TTEO with a boiling point below 40 °C will volatilize, stimulating the presence of pores in the membrane [46].

For formulation F4, a more homogeneous surface was observed compared to the other formulations. This observation suggests that ZnO-NPs and TTEO act synergistically despite the incompatibility observed for formulations F2 and F3 due to fragmentations on the membrane. Moreover, the TTEO increases the ZnO-NPs dispersion, facilitating the removal of -OH groups on the ZnO surface, causing a decrease in the value of the specific surface area of ZnO [47].

### 3.7. In Vivo Biocompatibility Tests of the PCL/PLA/ZnO-NPs/Gly/TTEO Nanocomposites

Before collecting the implanted samples on the biomodel tissues, macroscopic observation of the implanted area was performed following the UNE-EN ISO 10993-6:2017 standard entitled “biological evaluation of medical devices, part 6: Tests related to local effects after implantation”.

Figure 10 corresponds to the dorsal subdermal section of one of the biomodels. For all the biomodels, a similar appearance was observed after 60 days of implantation, finding in all the biomodels total recovery of the hair without purulent exudate. Subsequently, when staining tests were performed, it was observed that the surgical lesions healed entirely without necrotic processes. Additionally, the presence of fibrous capsules (encapsulation of the material) indicates a normal healing process with foreign bodies (membranes) fixing the tissue at the implantation site until its subsequent removal, which is in agreement with research on materials implanted on soft tissue.

#### 3.7.1. Histology Results for F1 (25%PCL/70%PLA/5%Gly)

The results of the histological analysis show that 60 days after the implantation of the F1 samples, there are many particles of variable size in the implantation area, surrounded by a fibrous capsule (Figure 11A). Growth of connective tissue in the form of a septum is also observed in the implantation area, which seems to delimit the material made up of bundles of collagen type III according to the Gomori staining technique (GT) (Figure 11B). At 100× magnification, it is observed that these septa contain inflammatory cells and large blood vessels.

The formation of a fibrous capsule surrounding the implantation area has already been reported for implanted materials. It corresponds to how the scarring process tries to limit the damage to allow the restoration of the lost tissue to take place. When a biomaterial is implanted, a series of events generates a foreign body reaction-type scarring response. First, the injected material is surrounded by a fibrous capsule with giant cells [48], as shown in Figure 11C.

The implanted material F1 contains 70% PLA, considered a biocompatible and utterly biodegradable material [49]; however, the degradability will depend on the molecular weight and the degree of crystallinity [50]. On the other hand, PCL has also considered a biocompatible and bioresorbable material with great potential in tissue engineering [51]. Since biodegradability is a condition of interest in biomedical applications, the combination of PLA and PCL has been proposed to regulate biodegradability [52].

#### 3.7.2. Histology Results for F2 (22%PCL/70%PLA/5%Gly/3%ZnO-NPs)

The observations for the F2 formulation at 60 days of implantation are similar to those reported for the F1 formulation. A large number of different sized fragments of the material are observed (Figure 12A). At a magnification of 40×, the pieces are smaller than those observed for the F1 formulation and are also surrounded by fibers of collagen type III (Figure 12B). At a magnification of 100×, the septa covering the portions of the material are smaller and have some inflammatory cells and blood vessels (Figure 12C).

The addition of 3 wt.% of ZnO-NPs in F2 improves the biodegradability and biocompatibility of the implanted product. In Figure 12, the fibrous capsule is no longer noticeable, as reported for formulation F1, and there seems to be greater penetration of connective tissue inside the implanted area with large blood vessels. In addition, the material is more fragmented because ZnO-NPs are potential activators of angiogenesis [53]. Furthermore, it has been reported that zinc oxide nanoparticles have an anti-inflammatory effect and are tissue-healing accelerators [54], explaining the more significant resorption of the F2 material compared to F1 during 60 days of subdermal implantation.

On the other hand, it is interesting to note that a slight modification in the proportions, decreasing the amount of PCL and adding ZnO, influences the processes of reabsorption and degradation of the material because the presence of a fibrous capsule is not noticeable, generating a decrease in the inflammatory infiltrate, indicating that ZnO promotes phagocytic processes.

#### 3.7.3. Histology Results for F3 (15%PCL/70%PLA/5%Gly/10%TTEO)

The histological image of F3 after 60 days of implantation is similar to that described for F2, with connective tissue in the middle of the fragments and a fibrous capsule surrounding the implantation area (Figure 13A). Figure 13B shows pieces of different sizes in the degradation process and signs of erosion at the edges. The smaller fragments are included within a connective tissue formed by type III collagen fibers (Figure 13B). At 100× magnification (Figure 13C), one of the fragments in the connective tissue is observed, surrounded by inflammatory cells and blood vessels, in an evident degradation process. In this formulation, we added tea tree essential oil (10 wt.%). TTEO is also considered biocompatible [55], with an antimicrobial effect [56], and the decrease in the amount of PCL (15 wt.%), which favored the process of resorption and degradation of the material, as evidenced by the abundant presence of type III collagen fibers, the presence of inflammatory cells and blood vessels, without the presence of an aggressive immune response. In our group, we have previously demonstrated the positive and accelerating effect of tea tree essential oil (TTEO) on the resorption of chitosan (CS) and polyvinyl alcohol (PVA) based biomaterials in subdermal tissue of Wistar rats [23].

#### 3.7.4. Histology Results for F4 (12%PCL/70%PLA/5%Gly/3%ZnO-NPs/10%TTEO)

At 60 days after implantation, the F4 material is observed to be more fragmented than the other formulations, with connective tissue in the middle of the fragments. The fibrous capsule is also observed (Figure 14A,B), with type III collagen both in the capsule and connective tissue according to GT staining (Figure 14B). Figure 14C shows several fragments in the connective tissue in an evident process of fragmentation and resorption. A tiny portion is also surrounded by inflammatory cells to continue its resorption process (area of the yellow circle in Figure 14C) and several blood vessels.

When a biomaterial is implanted, the organism will respond to the presence of a foreign body by encapsulating the inserted material. This reaction fades away when the material is wholly reabsorbed for its subsequent elimination. In stable materials, the material’s degradation rate takes longer, causing the appearance of inflammatory infiltrate. This process has been reported to indicate normal healing and tissue recovery [57,58,59]. However, the prolongation of inflammatory cells produced by traumatism generates an alteration in healing, making lacerations, ulcers, or pus [60].

Although fibrous encapsulation has been reported to be a usual response to the implantation of materials, the scarring process that occurred with the material studied presents some differences from that described in the literature since the fibrous capsule is described as a dense layer of parallel, avascular fibers [61,62]. However, the results of this work for the four formulations indicate that there is a connective tissue structure, similar to a fibrous capsule surrounding the material in the implantation zone, but with a high content of blood vessels (Figure 11, Figure 12, Figure 13 and Figure 14); additionally, the connective tissue projected towards the interior of the material surrounding portions of the material and contributing to its resorption by inflammatory cells.

The results of this work show that blood vessels and type III collagen fibers were present in all formulations. Angiogenesis is a permanent condition in the healing process. Moreover, blood vessels are indispensable for the four stages in which healing develops (hemostasis, inflammation, proliferation, and remodeling) [63], as well as type III collagen, which is present early in the replacement of the provisional matrix by the permanent matrix and in the constitution of other fibers in the healing process by interacting with type I collagen and different types of collagens [64].

In this sense, it is evident that the incorporation of the TTEO and ZnO-NPs accelerates the degradation process of the PLA/PCL polymeric matrix. This degradation seems to occur through the mechanism of fragmentation of the larger particles with the presence of connective tissue constituted by type III collagen fibers, blood vessels, and inflammatory cells, where the process of resorption of the implanted material continues.

## 4. Conclusions

In this work, four PCL/PLA/ZnO-NPs/Gly/TTEO membranes were synthesized that exhibited increased thermal stability concerning their pure components. Additionally, it was observed that adding ZnO-NPs favored the degradation and dominated the thermostable properties of TTEO, as shown for formulation F4. On the other hand, the surface architecture shows how the presence of TTEO or ZnO-NPs generates in the composites a rough drop-matrix type morphology due to the incompatibility between the polymers. However, it was elucidated that the presence of both components, TTEO and ZnO-NPs, in the F4 formulation act synergistically, showing homogeneous surfaces. On the other hand, the TTEO and ZnO-NPs’ presence was demonstrated by the characteristic bands for each component. In addition, the C-O-C band broadening was also attributed to intermolecular hydrogen bonding between the TTEO and the polymeric matrix, which was observed mainly in the F3 and F4 formulations. The PCL/PLA/ZnO-NPs/Gly/TTEO nanocomposites showed biocompatible in the subdermal implantation model at 60 days in the biomodels, allowing a healing process to occur with the recovery of hair and tissue architecture without the presence of an aggressive immune response. The material is fragmented and reabsorbed by inflammatory cells in the resolution process. Simultaneously the implantation zone is replaced by connective tissue with type III collagen fibers, blood vessels, and some inflammatory cells that continue the reabsorption process.

## Figures and Tables

**Figure 1 pharmaceutics-15-00043-f001:**
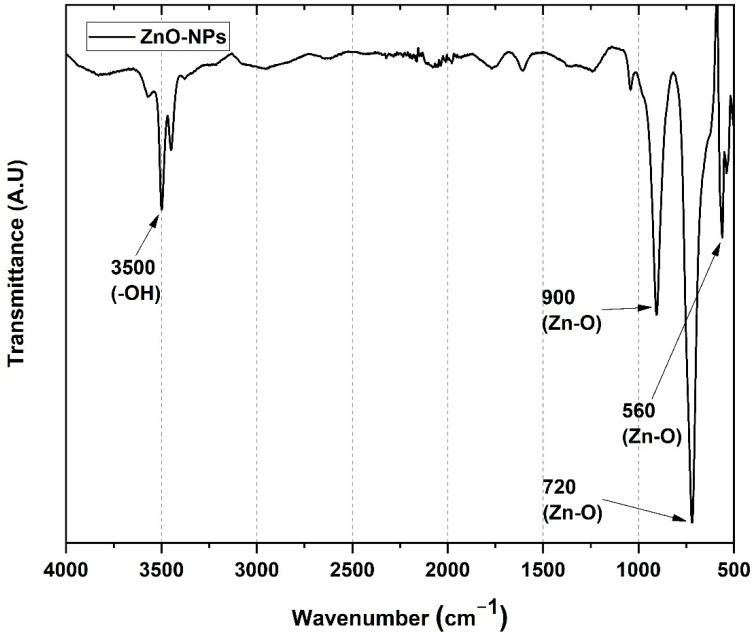
FT−IR spectrum for the ZnO-NPs.

**Figure 2 pharmaceutics-15-00043-f002:**
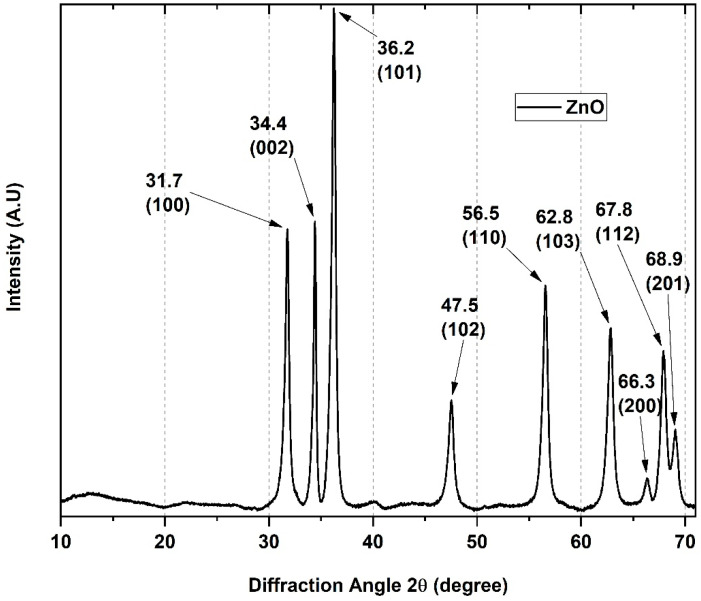
XRD patterns of the ZnO-NPs.

**Figure 3 pharmaceutics-15-00043-f003:**
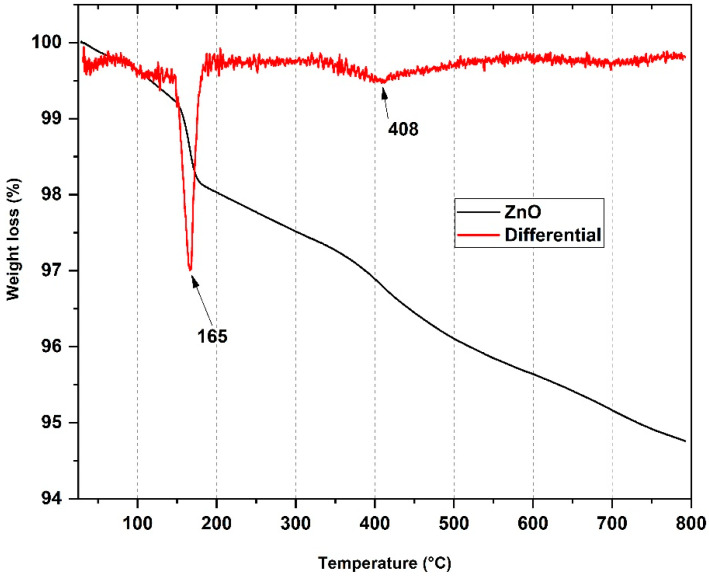
Thermogravimetric analysis of the ZnO-NPs.

**Figure 4 pharmaceutics-15-00043-f004:**
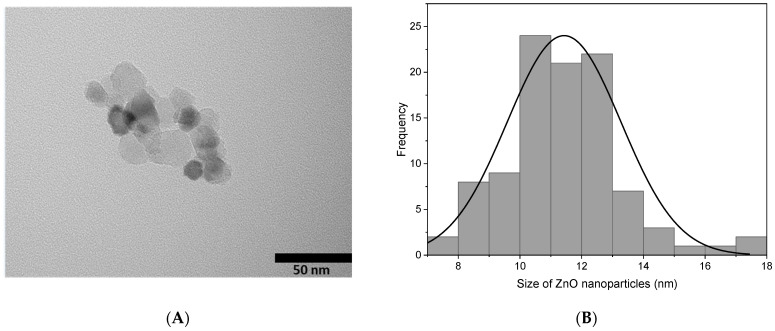
(**A**) TEM images of the ZnO-NPs at 50 nm and (**B**) histogram of the ZnO-NPs.

**Figure 5 pharmaceutics-15-00043-f005:**
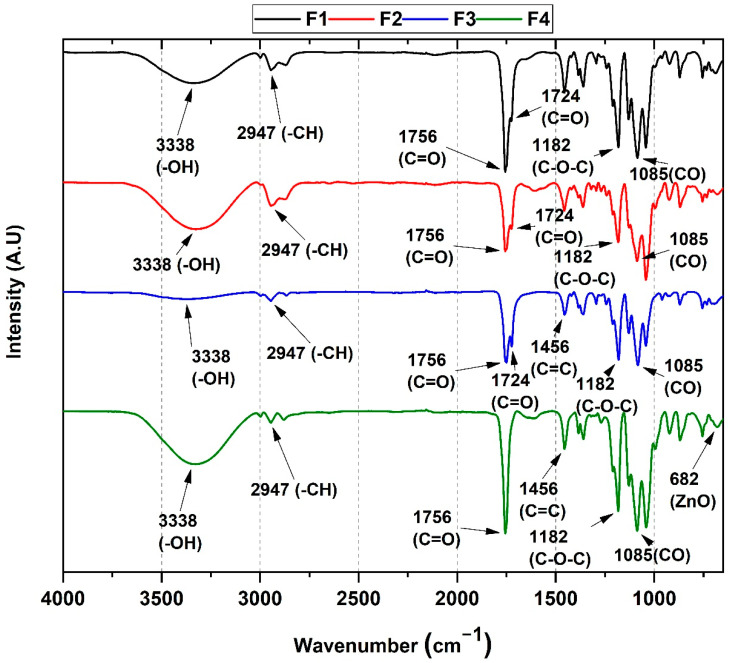
FT-IR spectrum of PCL/PLA/ZnO-NPs/Gly/TTEO nanocomposites. F1, 25%PCL/70%PLA/5%Gly; F2, 22%PCL/70%PLA/5%Gly/3%ZnO-NPs; F3, 15%PCL/70%PLA/5%Gly/10%TTEO; F4, 17%PCL/70%PLA/5%Gly/3%ZnO-NPs/10%TTEO.

**Figure 6 pharmaceutics-15-00043-f006:**
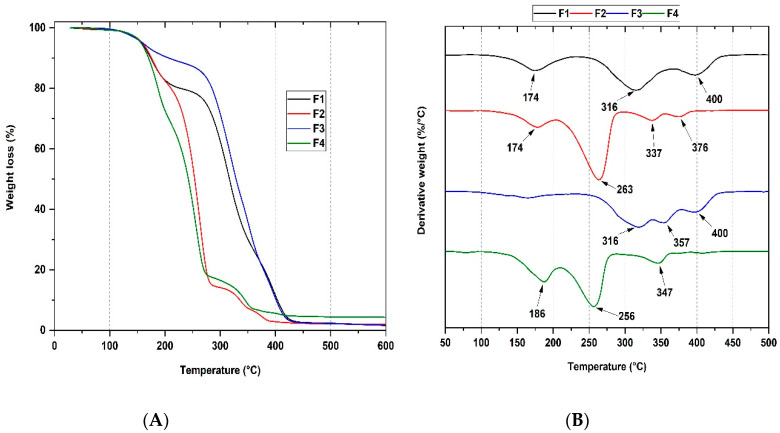
Thermogravimetric analysis of the PCL/PLA/ZnO-NPs/Gly/TTEO nanocomposites. F1, 25%PCL/70%PLA/5%Gly; F2, 22%PCL/70%PLA/5%Gly/3%ZnO-NPs; F3, 15%PCL/70%PLA/5%Gly/10%TTEO; F4, 17%PCL/70%PLA/5%Gly/3%ZnO-NPs/10%TTEO. (**A**): thermogram (TGA) of nanocomposites F1, F2, F3, and F4. (**B**): Derivative thermogravimetric analysis (DTGA) for the F1, F2, F3, and F4 nanocomposites.

**Figure 7 pharmaceutics-15-00043-f007:**
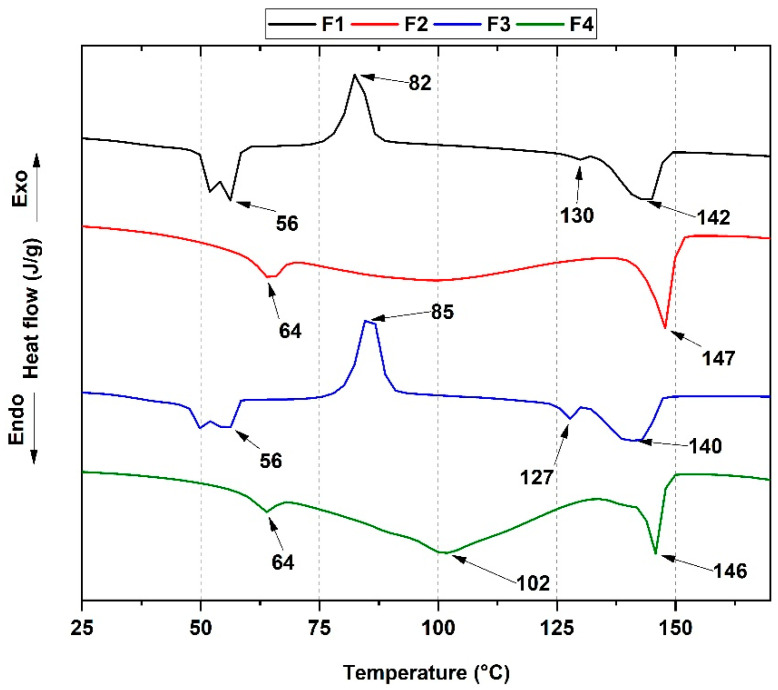
DSC curves of the PCL/PLA/ZnO-NPs/Gly/TTEO nanocomposites. Formulations: F1, 25%PCL/70%PLA/5%Gly; F2, 22%PCL/70%PLA/5%Gly/3%ZnO-NPs; F3, 15%PCL/70%PLA/5%Gly/10%TTEO; F4, 17%PCL/70%PLA/5%Gly/3%ZnO-NPs/10%TTEO.

**Figure 8 pharmaceutics-15-00043-f008:**
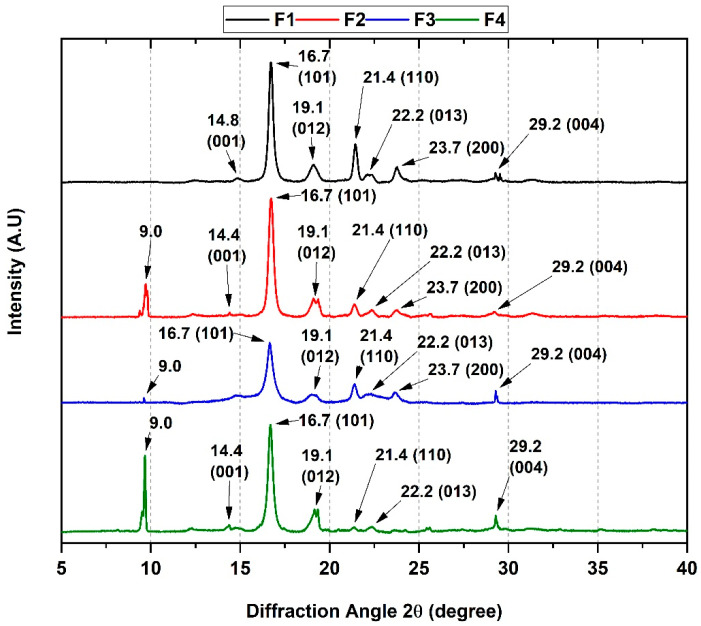
XRD analysis of PCL/PLA/ZnO-NPs/Gly/TTEO nanocomposites. F1, 25%PCL/70%PLA/5%Gly; F2, 22%PCL/70%PLA/5%Gly/3%ZnO-NPs; F3, 15%PCL/70%PLA/5%Gly/10%TTEO; F4, 17%PCL/70%PLA/5%Gly/3%ZnO-NPs/10%TTEO.

**Figure 9 pharmaceutics-15-00043-f009:**
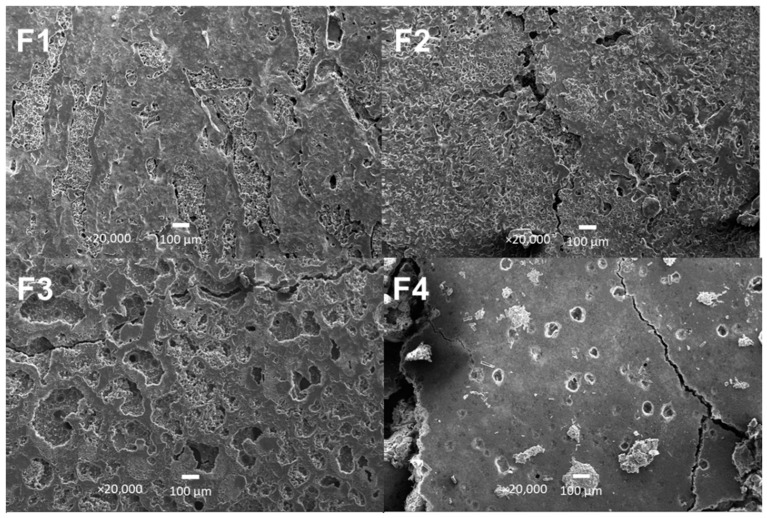
Morphology of PCL/PLA/ZnO-NPs/Gly/TTEO nanocomposites by SEM. F1, 25%PCL/70%PLA/5%Gly at 20,000×; F2, 22%PCL/70%PLA/5%Gly/3%ZnO-NPs at 20,000×; F3, 15%PCL/70%PLA/5%Gly/10%TTEO at 20,000×; F4, 17%PCL/70%PLA/5%Gly/3%ZnO-NPs/10%TTEO at 20,000×.

**Figure 10 pharmaceutics-15-00043-f010:**
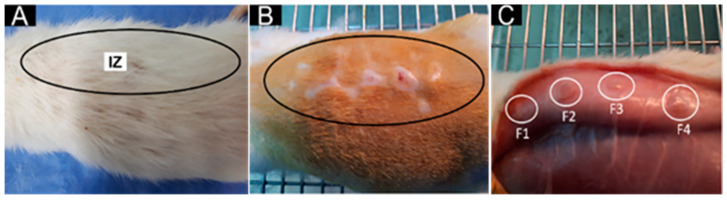
Subdermal dorsal section implanted in biomodel. (**A**): Dorsal area with hair. (**B**): Dorsal area with a trichotomy. (**C**): The internal area of the skin. Black ovals and IZ: implantation zone. White circles: individual implantation zones.

**Figure 11 pharmaceutics-15-00043-f011:**
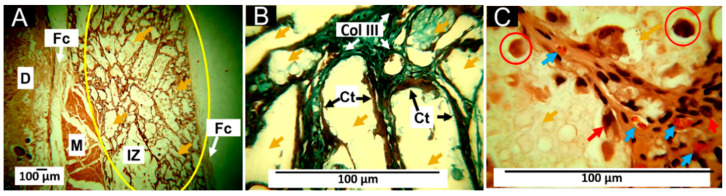
Subdermal implantations of F1 (25%PCL/70%PLA/5%Gly) at 60 days. (**A**): Formulation F1 at 4× HE technique. (**B**): Formulation F1 at 40× GT technique. (**C**): Formulation F1 at 100× HE technique. D: Dermis. M: Muscle. Fc: Fibrous capsule. IZ: Implantation zone. Yellow oval: Implantation area. Col III: Collagen type III. Ct: Connective tissue. Yellow arrows: Fragments of material F1. Red arrows: inflammatory cells. Blue arrows: blood vessels. Red circles: multinucleated cells. HE: Hematoxylin and Eosin technique. GT: Gomori Technique.

**Figure 12 pharmaceutics-15-00043-f012:**
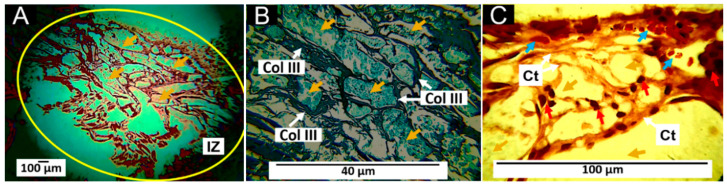
Subdermal implantations of F2 (22%PCL/70%PLA/5%Gly/3%ZnO-NPs) at 60 days. (**A**): Formulation F2 at 4× HE technique. (**B**): Formulation F2 at 40× GT technique. (**C**): Formulation F2 at 100× HE technique. Yellow oval: Implantation area. Col III: type III collagen. Ct: connective tissue. IZ: Implantation zone. Yellow arrows: Fragments of the F2 material. Red arrows: inflammatory cells. Blue arrows: blood vessels. HE: Hematoxylin and Eosin technique. GT: Gomori Technique.

**Figure 13 pharmaceutics-15-00043-f013:**
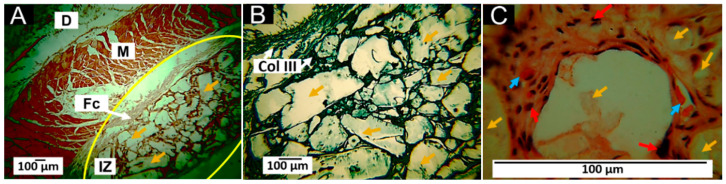
Subdermal implants of F3 (15%PCL/70%PLA/5%Gly/10%TTEO) at 60 days. (**A**): Formulation F3 at 4× HE technique. (**B**): Formulation F3 at 10× GT technique. (**C**): Formulation F3 at 100× HE technique. Yellow oval: Implantation zone. D: Dermis. M: Muscle. Fc: Fibrous capsule. IZ: Implantation zone. Col III: type III collagen. Yellow arrows: Fragments of material F3. Red arrows: Inflammatory cells. Blue arrows: Blood vessels. HE: Hematoxylin and Eosin technique. GT: Gomori Technique.

**Figure 14 pharmaceutics-15-00043-f014:**
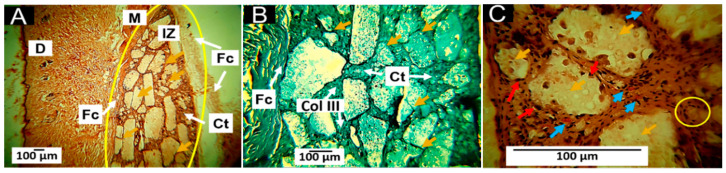
Subdermal implants of F4 (12%PCL/70%PLA/5%Gly/3%ZnO-NPs/10%TTEO) at 60 days. (**A**): Formulation F4 at 4× HE technique. (**B**): Formulation F4 at 10× GT technique. (**C**): Formulation F4 at 40× HE technique. Yellow oval: Implantation zone. D: Dermis. M: Muscle. Fc: fibrous capsule. Ct: connective tissue. IZ: Implantation zone. Col III: type III collagen. Yellow arrows: Fragments of material F3. Red arrows: Inflammatory cells. Blue arrows: Blood vessels. Yellow circle: Inflammatory cells surrounding a small fragment.

**Table 1 pharmaceutics-15-00043-t001:** Percentage composition (wt.%) for each component in the PCL/PLA/ZnO-NPs/Gly/TTEO membranes.

Components	F1	F2	F3	F4
PCL (%)	25	22	15	12
PLA (%)	70	70	70	70
Gly (%)	5	5	5	5
ZnO-NPs (%)	0	3	0	3
TTEO (%)	0	0	10	10

**Table 2 pharmaceutics-15-00043-t002:** Thermal properties of membranes PCL/PLA/ZnO-NPs/Gly/TTEO. F1, 25%PCL/70%PLA/5%Gly; F2, 22%PCL/70%PLA/5%Gly/3%ZnO-NPs; F3, 15%PCL/70%PLA/5%Gly/10%TTEO; F4, 17%PCL/70%PLA/5%Gly/3%ZnO-NPs/10%TTEO.

	T_g_ (°C)	T_cc_ (°C)	T_m1_ (°C)	T_m2_ (°C)	T_m3_ (°C)	Xc PLA (%)
**F1**	55	82	130	142	56	5.1
**F2**	57	-	-	147	64	-
**F3**	55	85	127	140	56	4.7
**F4**	52	-	-	146	64	-

## Data Availability

Data will be available under request from the corresponding author.

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
