# Peer review of "Polycaprolactone (PCL)-Polylactic Acid (PLA)-Glycerol (Gly) Composites Incorporated with Zinc Oxide Nanoparticles (ZnO-NPs) and Tea Tree Essential Oil (TTEO) for Tissue Engineering Applications"

_pharmaceutics, 2022, doi:10.3390/pharmaceutics15010043_

Round 1

Reviewer 1 Report

Dear Authors, in your interesting manuscript, the following points should be added/changed to further improve it:

1.       Abstract: Please round the average size of ZnO NPs to whole numbers.

2.       Introduction: Please add information that zinc oxide is a multifunctional material.

3.       Synthesis of ZnO-NPs: Please add information about the ultrasonic bath (model, manufacturer).

4.       Synthesis of ZnO-NPs: Please add information about the calcination furnace (model, manufacturer).

5.       Synthesis of ZnO-NPs: Please explain to me the point of quoting reference number 24 reporting results about TiO2?

6.       Table 2. Please add information specifying the percentage (by weight or volume).

7.       X-Ray Diffraction (XRD) of ZnO-NPs: I have a comment on the sentence  “There are evident angular displacements 2theta: 31, 34, 254 36, 48, 56, 63, 66, 68, and 69° according to the (100), (002), (101), (102), (110), (103), (200), 255 (112) and (201) [9].”  I suggest that the authors refer to the XRD results for the JCPDS reference sample number 36-1451 (DOI:10.1155/2016/2789871) and provide the 2theta values with at least one decimal point.

8.       X-Ray Diffraction (XRD) of ZnO-NPs: I suggest calculating the average value of ZnO crystallites on the basis of the XRD results obtained.

9.       Transmission Electron Microscopy (TEM): Please round the average size of ZnO NPs to whole numbers.

10.   Transmission Electron Microscopy (TEM): There is a lack of discussion. Please compare your results with those obtained in the cited paper [23] and add discussions.

11.   Figure 9. Figure 9 does not match its description. SEM images are not named (A, B ,C .... H). Please correct Figure 9, add the missing SEM images, and add the appropriate descriptions on the SEM images.

12.   Please add information about the average size of the obtained ZnO particles.

Author Response

We are deeply thankful for the valuable suggestions and corrections to improve the quality of our manuscript. The corrections are presented below point by point in red for easy comprehension.

Reviewer 1

  1. Abstract: Please round the average size of ZnO NPs to whole numbers.

R// Thank you for your appreciation. Line 26 has a particle size rounded to 11 nm ± 2 nm.

The sol-gel method was used for zinc oxide nanoparticle synthesis with an average size of 11 nm ± 2 nm and spherical morphology

  1. Introduction: Please add information that zinc oxide is a multifunctional material.

R// The correction can be found in line 53 as follows:

Zinc oxide nanoparticles (ZnO-NPs) a multifunctional material recognized as safe materials used as a food additive considered GRAS and approved by the Food and Drug Administration (FDA)

  1. Synthesis of ZnO-NPs: Please add information about the ultrasonic bath (model, manufacturer).

R// We appreciate the suggestion. The information added is in line 106:

Then, the nanoparticles were dispersed in an ultrasonication bath (Branson, Madrid, Spain) and mixed with 2-propanol for 10 min at 20 °C.

  1. Synthesis of ZnO-NPs: Please add information about the calcination furnace (model, manufacturer).

R// We appreciate the suggestion. The oven description was added in line 110: Finally, the nanoparticles were calcined in an oven Nabertherm LHT 02/18 (Lilienthal, Bremen, Germany) at 250 °C for five hours."

  1. Synthesis of ZnO-NPs: Please explain to me the point of quoting reference number 24 reporting results about TiO2?

R// We appreciate the observation. We corrected the information to "The synthesis of the nanoparticles was carried out according to the procedure already reported [24]," using the following reference:

  1. Becheri, A.; Dürr, M.; Lo Nostro, P.; Baglioni, P. Synthesis and characterization of zinc oxide nanoparticles: application to textiles as UV-absorbers. J. Nanoparticle Res. 2008, 10, 679–689.
  2. Table 2. Please add information specifying the percentage (by weight or volume).

R// We appreciate the recommendation. The information was added in line 125 as weight percentage (table 1).

Table 1. Percentage composition (wt. %) for each component in the PCL/PLA/ZnO-NPs/Gly/TTEO membranes.

  1. X-Ray Diffraction (XRD) of ZnO-NPs: I have a comment on the sentence  "There are evident angular displacements 2theta: 31, 34, 254 36, 48, 56, 63, 66, 68, and 69° according to the (100), (002), (101), (102), (110), (103), (200), 255 (112) and (201) [9]."  I suggest that the authors refer to the XRD results for the JCPDS reference sample number 36-1451 (DOI:10.1155/2016/2789871) and provide the 2theta values with at least one decimal point.

R// Thank you very much for the suggestion. The 2θ displacements were corrected with one decimal.

There are evident angular displacements 2θ: 31.7, 34.4, 36.2, 47.5, 56.5, 62.8, 66.3, 67.8, and 68.9° according to the (100), (002), (101), (102), (110), (103), (200), (112) and (201)

  1. X-Ray Diffraction (XRD) of ZnO-NPs: I suggest calculating the average value of ZnO crystallites on the basis of the XRD results obtained.

R// Thank you very much for the observation. You can find the calculation of the average crystallite size corresponding to ZnO-NPs.

The average crystallite sizes (τ) for the ZnO-NPs were determined using Debye-Scherrer equation 2. 

2

Where K is Scherrer constant, and the crystalline shape factor is 0.89, λ represents the wavelength of X-ray source 1.5405 Å used in XRD, β is full width at half maximum of diffraction peak, and θ is the Bragg angle of the intense peak. In this regard, from the diffractogram data of the ZnO-NPs, the average crystallite size calculated is 30.13 nm.

  1. Transmission Electron Microscopy (TEM): Please round the average size of ZnO NPs to whole numbers.

R// Thank you very much for the observation. Within the text, it can be found as follows (lines 26 and 299):

Spherical nanoparticles have an average size of 11 nm ± 2 nm.

  1. Transmission Electron Microscopy (TEM): There is a lack of discussion. Please compare your results with those obtained in the cited paper [23] and add discussions.

R// Thank you very much for the suggestion. We added more information and discussion comparing the results with those of reference 23 between lines 292-300.

Figure 4 shows the morphology and size of the ZnO-NPs by transmission electron microscopy (TEM). Nearly spherical and fairly monodisperse nanoparticles are observed. However, some aggregates are found because when dry nanoparticle powder is added to water, the nanoparticles have a high local concentration with high surface energy, causing a high collision frequency and high aggregation [1,2]. The average diameter of the nanoparticles was found by averaging the size of 100 particles using Image J software. Considering the processed dimensions, we made a histogram that relates the frequency to the particle distribution (figure 4b). In this sense, the diameter of the nanoparticles was 11 nm ± 2 nm.

  1. Figure 9. Figure 9 does not match its description. SEM images are not named (A, B ,C .... H). Please correct Figure 9, add the missing SEM images, and add the appropriate descriptions on the SEM images.

R// Thank you very much for the correction. We corrected the text where you can find the corresponding images for the different formulations at 20000 x.

Figure 9. Morphology of PCL/PLA/ZnO-NPs/Gly/TTEO nanocomposites by SEM. F1, 25%PCL/70%PLA/5%Gly (F1) at 20000 ×; 22%PCL/70%PLA/5%Gly/3%ZnO-NPs (F2) at 20000 ×; 15%PCL/70%PLA/5%Gly/10%TTEO (F3) at 20000 ×; 17%PCL/70%PLA/5%Gly/3%ZnO-NPs/10%TTEO (F4) at 20000 ×

  1. Please add information about the average size of the obtained ZnO particles.

R// The average size of the ZnO nanoparticles can be found in lines 26 and 299.

Reviewer 2 Report

I went through the manuscript of Grande-Tovar and co-authors and I found this a robust study of characterization which deserves publication on Pharmaceutics, besides a couple of minor concerns I mandatory recommend to fix before resubmission to this journal. 

1) typo at line 96

2) Fig 4 reports unnecessary repetition of the same TEM image: one is enough.

3) Do the authors have evidence of how the ZnO-NPs size influences the degradation process of the LA/PCL polymeric matrix?

4) Lines 435-437 Sentence too long. Please split it and explain better. 

5) The wide body of works on PCL is still ongoing with various application fields, so more recent works or reviews on this subject worths mention in the main text. A couple of examples: 

https://doi.org/10.3390/biology10050398

https://doi.org/10.1021/acsabm.2c00174

https://doi.org/10.1016/j.jobcr.2019.10.003

Author Response

We are deeply thankful for all the valuable suggestions to improve the quality of our manuscript. The corrections are presented point by point for easy comprehension. 

Reviewer 2

  1. Typo at line 96

R// Thank you very much for the suggestion. We had a mistake since the nanoparticles were not organically modified and removed that expression.

  1. Fig 4 reports unnecessary repetition of the same TEM image: one is enough.

R// Thank you very much for the observation. The TEM image at 50 nm was the only figure reported.

(A)

(B)

Figure 4. (A) TEM images of the ZnO-NPs at 50 nm and (B) histogram of the ZnO-NPs

  1. Do the authors have evidence of how the ZnO-NPs size influences the degradation process of the LA/PCL polymeric matrix?

R// Thank you very much for the observation. We do not have any evidence of the degradation produced by the zinc oxide nanoparticles on the PLA/PCL matrix. However, some studies have shown how different nanoparticles (TiO2, ZnO, among others) promote polymer degradation, decreasing in many cases the thermal and mechanical properties observed in the results of TGA and young models, respectively. In addition, some SEM images of these studies show how the nanoparticles promote degradation causing islands or porosity to the material [3–5].

  1. Lines 435-437 Sentence too long. Please split it and explain better.

R// Thank you very much for the suggestion. The sentence was split into two sentences.

The irregularities observed in F3 as islands and microvoids are mainly due to the hydrophobic character of the lipids present in the TTEO, which causes flocculation of the components above the polymeric matrix. On the other hand, when the mixture is subjected to 40 °C in the preheated oven for membrane formation, the natural components of the TTEO with a boiling point below 40 °C will volatilize, stimulating the presence of pores in the membrane [45].

  1. The wide body of works on PCL is still ongoing with various application fields, so more recent works or reviews on this subject worths mention in the main text. A couple of examples: 

https://doi.org/10.3390/biology10050398

https://doi.org/10.1016/j.jobcr.2019.10.003

            https://doi.org/10.1021/acsabm.2c00174

R// Thank you very much for the suggestion. The mentioned works were cited:

  1. Schmitt, P.R.; Dwyer, K.D.; Coulombe, K.L.K. Current Applications of Polycaprolactone as a Scaffold Material for Heart Regeneration. ACS Appl. Bio Mater. 2022.

51.Dwivedi, R.; Kumar, S.; Pandey, R.; Mahajan, A.; Nandana, D.; Katti, D.S.; Mehrotra, D. Polycaprolactone as biomaterial for bone scaffolds: Review of literature; Craniofacial Research Foundation, 2019; ISBN 9335902322.

60. Glasses, M.B.; Petretta, M.; Gambardella, A.; Boi, M.; Berni, M.; Cavallo, C.; Marchiori, G.; Maltarello, M.C.; Bellucci, D.; Fini, M.; et al. Composite Scaffolds for Bone Tissue Regeneration Based on. 2021, 1–18.

Reviewer 3 Report

The manuscript entitled “Polycaprolactone (PCL)-Polylactic acid (PLA)-Glycerol (Gly) Composites Incorporated with Zinc Oxide nanoparticles (ZnO- NPs) and Tea Tree Essential Oil (TTEO) for Tissue Engineering Applications” described the fabrication of composites out of Polycaprolactone (PCL)-Polylactic acid (PLA) Composites Incorporated with Zinc Oxide nanoparticles (ZnO- NPs) and Tea Tree Essential Oil (TTEO) using glycerol as a plasticizer for Tissue Engineering Applications. However, the manuscript suffers from a number of shortcomings as described below: 

1.     Line 49 has no meaning in the sentence. Alter the sentence so that the readers can easily understand (Page 1)

2.     The letter I should be in capital in case of Image J software (Line 133, Page 3)

3.     The authors should note that it is 2θ instead of 2q (Line 137, Page 3)

4.     Line 178 is very unclear with numbers such as 93 in it. Authors should kindly check it (Page 4)

5.     In Figure 5 -F2 please put an arrow mark for 1182cm-1  and also make sure to have uniform arrow marks throughout the graph (Page 9)

6.     The authors have to give separate description for Figures 6A and 6B (Page 9)

7.     The miller indices (hkl) values have to be put in brackets (Line 409, 410) and also has to be mentioned in the XRD plot (Figure 8) (Page 12)

8.     The unit of the x-axis in figure 8 should be mentioned as degree and not as deg (Page 12)

9.     The authors should also mention what is IZ (Figure 11A) (Page 14)

10.  The authors should also mention about Figure 11C, Figure 12C and Figure 13C in text and if mentioned put the figure number in brackets. (Page 14,15,16)

Author Response

We are deeply thankful for all the valuable suggestions to improve the quality of our manuscript. The corrections are presented point by point for easy comprehension. 

  1. Line 49 has no meaning in the sentence. Alter the sentence so that the readers can easily understand (Page 1)

R// Thank you very much for your appreciation. The sentence has been corrected for better understanding.

However, these polymers present an intrinsic stiffness of the carbonated chain, which becomes a disadvantage due to their low tensile elongation and toughness properties that limit their industrial applications

  1. The letter I should be in capital in case of Image J software (Line 133, Page 3)

R// Thank you very much for the correction, which can be founded in Line 145.

The average of 100 zinc oxide nanoparticles was considered to determine the particle size by processing in Image J 1.49q software.

  1. The authors should note that it is 2θ instead of 2q (Line 137, Page 3)

R// Thank you very much for the observation. We corrected the text in Line 149.

The crystalline structure was evaluated using a PANalytical X0Pert PRO diffractometer (Malvern Panalytical, Jarman Way, Royston, UK) using copper radiation with a wavelength of Kα1 (1.540598 Å) and Kα2 (1.544426 Å) operated in the secondary electron mode at 45 kV in a 2θ range between 5 and 70°.

  1. Line 178 is very unclear with numbers such as 93 in it. Authors should kindly check it (Page 4)

R// Thank you very much for the observation. It has been corrected in the text. Line 190

Where  correspond to the theoretical enthalpy of the completely crystalline polymer, which is 93 J/g for PLA

  1. In Figure 5 -F2 please put an arrow mark for 1182cm-1and also make sure to have uniform arrow marks throughout the graph (Page 9)

R// Thank you very much for the suggestion. The FTIR graph was corrected.

  1. The authors have to give separate description for Figures 6A and 6B (Page 9)

R// Thank you very much for the observation. The description has been corrected for the thermogravimetric analysis and the derivative of the degradation percentage concerning temperature.

Figure 6. Thermogravimetric analysis of the PCL/PLA/ZnO-NPs/Gly/TTEO nanocomposites. F1, 25%PCL/70%PLA/5%Gly; F2, 22%PCL/70%PLA/5%Gly/3%ZnO-NPs; F3, 15%PCL/70%PLA/5%Gly/10%TTEO; F4, 17%PCL/70%PLA/5%Gly/3%ZnO-NPs/10%TTEO. A: thermogram (TGA) of nanocompo-sites F1, F2, F3, and F4. B: Derivative thermogravimetric analysis (DTGA) for the F1, F2, F3, and F4 nanocomposites.    

  1. The miller indices (hkl) values have to be put in brackets (Line 409, 410) and also has to be mentioned in the XRD plot (Figure 8) (Page 12)

R// Thank you very much for the comment. The graph was corrected.

Planes 2θ at 14.8, 16.7, 19.1, 22.2, and 29.2° attributed to planes (001), (101), (012), (013), and (004), respectively. It was also possible to observe the diffraction peaks of orthorhombic PCL at 21.4 and 23.7°°, corresponding to the (110) and (200) planes.

  1. The unit of the x-axis in figure 8 should be mentioned as degree and not as deg (Page 12)

R// Thank you very much for the comment. The graph was corrected.

  1. The authors should also mention what is IZ (Figure 11A) (Page 14)

R// Thank you very much for the suggestion. The graph shows black circles as the acronym IZ, which means implantation zone- We added the explanation to figures 10-14.

  1. The authors should also mention about Figure 11C, Figure 12C, and Figure 13C in text and if mentioned put the figure number in brackets. (Page 14,15,16)

R// Thank you very much for the suggestion. We mentioned the figures as follows:

First, the injected material is surrounded by a fibrous capsule with giant cells [48], as shown in Figure 11C. Line 520-521.

At a magnification of 100×, the septa covering the portions of the material are smaller and have some inflammatory cells and blood vessels (Figure 12C). Line 534-535.

At 100× magnification (Figure 13 C), one of the fragments in the connective tissue is observed, surrounded by inflammatory cells and blood vessels, in an evident degradation process. Line 563-565.

Reviewer 4 Report

The manuscript entitled „Polycaprolactone (PCL)-Polylactic acid (PLA)-Glycerol (Gly) Composites Incorporated with Zinc Oxide nanoparticles (ZnO-NPs) and Tea Tree Essential Oil (TTEO) for Tissue Engineering Applications”, submitted for evaluation to Pharmaceutics, presents the results of characterization of structure and in vivo biodegradation (in histological evaluation) of complex polymer materials. In particular, effect of ZnO NPs and essential oil of tea tree is taken into consideration.

Evaluation of the properties of the mentioned composites does not even include the basic elements of assessing antibacterial activity, although both additives have confirmed antibacterial activity. Nevertheless, the presented results contribute to the field of biomaterials. In this context, I consider the topic of the work to be some importance.

 In general, the work is clear. It is also written in the correct language. However, I have some comments and I found some errors. My comments and questions concerning the submitted article are listed below:

 COMMENTS TO AUTHORS

  1. Section 2.3.: please provide information on the origin or method of isolation of TTEO used in this study.
  2. Section 2.4.: There is no clear information on the composition of the membranes. do the authors mean by "4% of total solids" the content of PLA, PCL and ZnO exclusively? Glycerol and TTEO are liquids… Please provide more clear description of membrane composition.

3.     Section 2.6., surgical preparation of biomodels: It was stated that “Then, four 5 mm incisions were made on the right side of the midline, and three pockets 1 cm wide and 10 cm deep were created with hemostatic forceps, with a centimeter of separation between them [27].”. Is it correct that depth of created pockets was 10 cm (100 mm)? Moreover, if four incisions but only 3 pockets were formed in each rat dorsal area, how it is possible that 4 membranes were implanted? It was confirmed by legend of Figure 10. I understand from the description, that only 3 rats were used in the entire in vivo study, so, 4 biomaterials were implanted into 3 animals (n=3).

  1. Figures 11-14 should be, in my opinion, combined into one. This will make it easier to compare all results for the 4 tested membranes.
  2. Figure 4 legend: Please rephrase the fragment “TEM images of the ZnO-NPs at 50 and 100 nm”, to clarify that 50 and 100 nm reefrs to scale bars.
  3. Figure 10: Descriptions of 4 individual images are missing (A, B etc) or are hardly visible on images. In addition, according to the legend, the figure should contain 8 images (A-H), but in reality it only contains 4 images. Moreover, I suggest to add arrows or circles to indicate the location of the structures mentioned in the text.
  4. Please inform why no evaluation of antimicrobial activity of created membranes was included in the plan of this study.

Author Response

We are thankful for all the valuable corrections to improve the quality of our manuscript. The corrections are presented below point by point in red for easy comprehension.

  1. Section 2.3.: please provide information on the origin or method of isolation of TTEO used in this study.

R// Thank you very much for the suggestion. The materials and methods section presents that the essential oil was purchased from Marnys company (Madrid, Spain) and how the essential oil was characterized (lines 115-122).

The essential tea oil (TTEO) was purchased from Marnys (Madrid, Spain), and its compositional analysis was characterized by gas chromatography coupled to mass spectrometry (GC-MS) using C6 to C25 hydrocarbons as reference. The gas chromatograph was an AT6890 series plus (Agilent Technologies, Palo Alto, CA, USA) coupled to a mass selective detector (Agilent Technologies, MSD 5975). Column DB-5MS (J & W Scientific, Folsom, CA, USA), 5% -Ph-PDMS. Identification comparing RI (retention indexes) with the Adams database (Wiley, 138 and NIST05, Agilent, Santa Clara, California, United States).

  1. Section 2.4.: There is no clear information on the composition of the membranes. do the authors mean by "4% of total solids" the content of PLA, PCL, and ZnO exclusively? Glycerol and TTEO are liquids… Please provide more clear description of membrane composition.

R// Thank you very much for your appreciation. The procedure was corrected as follows (lines 125-138):

For the preparation of the PCL/PLA/ZnO-NPs/Gly/TTEO membranes, the main parameter considered was the presence of a concentration final of 4% (wt.%), according to the wt.% ratio of each formulation. Then, a dispersion of ZnO-NPs (300 mg/10 mL) in chloroform was prepared using an ultrasonic bath (Branson, Madrid, Spain) for two hours. For the content of glycerol employed, its density of 1.26 g/cm3. Subsequently, each component was dissolved in chloroform and mixed according to Table 2. Finally, the resulting mixture was placed in an ultrasonic bath (Branson, Madrid, Spain) for two hours to eliminate the pre-existing bubbles in the solution.

  The mixture was poured into glass molds for 24 h and cured in a preheated oven at 40 °C ± 0.2 to obtain PCL/PLA/ZnO-NPs/Gly/TTEO membranes.

  1. Section 2.6., surgical preparation of biomodels: It was stated that "Then, four 5 mm incisions were made on the right side of the midline, and three pockets 1 cm wide and 10 cm deep were created with hemostatic forceps, with a centimeter of separation between them [27].". Is it correct that depth of created pockets was 10 cm (100 mm)? Moreover, if four incisions but only 3 pockets were formed in each rat dorsal area, how it is possible that 4 membranes were implanted? It was confirmed by legend of Figure 10. I understand from the description, that only 3 rats were used in the entire in vivo study, so, 4 biomaterials were implanted into 3 animals (n=3)

R// We appreciate the correction from the reviewer. We corrected the information between lines (209-213). The number of biomodels was determined following the ISO 10993-6 standard, which defines that three is the minimum number that should be used and respecting the principle of "reduction" that recommends using the minimum possible number of biomodels:

Then, the area to be implanted was determined (right side of the midline), where anesthesia was applied with an infiltrative technique using 2% Lidocaine with epinephrine (Newstetic, Guarne, Colombia). Then, four 5 mm incisions were made on the right side of the midline, and four pockets 10 mm wide and 10 mm deep were created with hemostatic forceps, with a centimeter of separation between them [27].

  1. Figures 11-14 should be, in my opinion, combined into one. This will make it easier to compare all results for the 4 tested membranes.

R// Thank you very much for your appreciation. However, we believe that the resolution of the different images will be lost if we combine them. The high resolution is essential to show where the connective tissue is located and the inflammatory infiltrates. Besides, the legend of each figure is extensive, and combining them might confuse the readers.

  1. Figure 4 legend: Please rephrase the fragment "TEM images of the ZnO-NPs at 50 and 100 nm" to clarify that 50 and 100 nm refers to scale bars.

R// Thank you very much for the correction. One referee suggested showing only one of the TEM images of the nanoparticles. Therefore, it was decided to put only the image at 50 nm.

TEM images of the ZnO-NPs at 50 nm

  1. Figure 10: Descriptions of 4 individual images are missing (A, B, etc) or are hardly visible on images. In addition, according to the legend, the figure should contain 8 images (A-H), but in reality it only contains 4 images. Moreover, I suggest to add arrows or circles to indicate the location of the structures mentioned in the text.

R// Thank you very much for the observation. Finally, the image of figure 10 is shown with its respective description.

Figure 10. Subdermal dorsal section implanted in biomodel. A: Dorsal area with hair. B: Dorsal area with a trichotomy. C: The internal area of the skin. Black ovals and IZ: implantation zone. White circles: individual implantation zones.

  1. Please inform why no evaluation of the antimicrobial activity of created membranes was included in the plan of this study.

R// For this study, we wanted to analyze these membranes' preliminary biocompatibility and biodegradability. Still, antimicrobial activity was not the focus of this study. Still, we will assess it soon with different strains of gram-positive and gram-negative bacteria and cytotoxic activity against cancer cells in a further investigation.

Round 2

Reviewer 3 Report

The authors have addressed the queries raised by the reviewers. 

Reviewer 4 Report

Thank you for all answers and corrections. I have no further questions.